# Zirconium Carbide for Hypersonic Applications, Opportunities and Challenges

**DOI:** 10.3390/ma16186158

**Published:** 2023-09-11

**Authors:** Glenn R. Peterson, Ryan E. Carr, Ernesto E. Marinero

**Affiliations:** School of Materials Engineering, Purdue University, West Lafayette, IN 47907, USA

**Keywords:** ZrC, hypersonics, ultra-high temperature ceramics

## Abstract

At ultra-high temperatures, resilient, durable, stable material choices are limited. While Carbon/Carbon (C/C) composites (carbon fibers and carbon matrix phases) are currently the materials of choice, zirconium carbide (ZrC) provides an option in hypersonic environments and specifically in wing leading edge (WLE) applications. ZrC also offers an ultra-high melting point (3825 K), robust mechanical properties, better thermal conductivity, and potentially better chemical stability and oxidation resistance than C/C composites. In this review, we discuss the mechanisms behind ZrC mechanical, thermal, and chemical properties and evaluate: (a) mechanical properties: flexure strength, fracture toughness, and elastic modulus; (b) thermal properties: coefficient of thermal expansion (CTE), thermal conductivity, and melting temperature; (c) chemical properties: thermodynamic stability and reaction kinetics of oxidation. For WLE applications, ZrC physical properties require further improvements. We note that materials or processing solutions to increase its relative density through sintering aids can have deleterious effects on oxidation resistance. Therefore, improvements of key ZrC properties for WLE applications must not compromise other functional properties. We suggest that C/C-ZrC composites offer an engineering solution to reduce density (weight) for aerospace applications, improve fracture toughness and the mechanical response, while addressing chemical stability and stoichiometric concerns. Recommendations for future work are also given.

## 1. Introduction

Zirconium carbide is an ultra-high temperature ceramic (UHTC) material, characterized by a very high melting point, high fracture toughness, and maintained strength at high temperatures. ZrC is promising for many aerospace and nuclear applications, specifically as a material to replace **^C/C^** composites as a Wing Leading Edge (WLE) material [1]. Vasile et al. have reported the utilization of ZrC composites as thermal barriers for the aerospace industry [2]. However, ZrC presents materials engineering challenges that must be solved for its practical implementation. One of these challenges is the fact that ZrC is a non-stoichiometric binary compound resulting from synthesis in materials with either a zirconium surplus or carbon deficiency [3]. A survey of the phase diagram of Zr-C, shown in Figure 1, is useful in understanding the reasons for its non-stoichiometric composition [4]. ZrC can be formed with a wide carbon composition ranging from 36 at% to 50 at%. Outside this range, multiphase compounds are formed. This wide range differs from other materials that might be described as line compounds and have very narrow stoichiometric ranges. This wide composition range in ZrC impacts, amongst other things, its melting point.

Across this wide composition range, the temperatures at which ZrC is stable vary over 1000 °C. At the lowest carbon concentration (36 at%), it is only stable up to 1854 °C; at this temperature, it undergoes a phase transition to a two-phase composition. A phase transition is also observed at the maximum carbon concentration (50 at%) at 2927 °C. Figure 1 shows that the most stable composition at high temperatures occurs at around C = 45%. At this composition, ZrC is stable until it melts at 3427 °C. The non-stoichiometric ratio of Zr:C leads to the creation of vacancies in the lattice. Impurities can occupy these vacancies, distorting lattice parameters, especially at high temperatures. This distortion results in decreased mechanical and thermal properties and impacts its chemical stability. For example, Tiwari and Feng note the strong influence of defects and impurities as factors that inhibit phonon vibration that limits attaining the theoretical thermal conductivity at ultra-high temperatures [5]. Conversely, defects in engineering could potentially be employed to fill these vacancies with judiciously chosen elements to yield materials with enhanced material properties. Most current research in ZrC examines its use in high-temperature composites, coatings, and ablative mechanisms on account of its strengthening properties, thermal conductivity, or CTE match to substrates [6,7,8,9,10,11,12].

In this review, potential pathways to achieve optimal thermal and mechanical properties of ZrC are also discussed. Zirconium carbide—a monolithic ceramic—has fewer compatibility concerns compared to other composites such as CTE mismatch with protective overcoats or chemical interaction concerns of multiphase composites, making the use of ZrC for WLE applications most promising. 

### 1.1. ZrC Physical Properties 

In this review, the physical properties of ZrC are discussed and compared to those of C/C composites. In addition to ultra-high melting points, another major advantage of C/C composites is their specific strength. Specific strength is the ratio of the material strength divided by its density. As shown in Figure 2, Advanced C/C, or ACC, are composites exhibiting the highest specific strength of any material at elevated temperatures [13]. 

While specific strength is an important consideration, other material properties need to be considered in evaluating ZrC suitability for WLE applications. These are summarized in Table 1 [11,13,15,16,17,18,19]. 

Table 1 lists values at room temperature for C/C and ZrC. The CTE of C/C is negative from room temperature to 400 °C; thereafter, it increases from 0.4 to 0.6 × 10^−6^ K^−1^ at 900 °C [20]. This is a challenge for protective coating solutions with materials with a different CTE. ZrC has a CTE that is closer matched to other materials at room temperature, increasing to 9.0 and 10.2 × 10^−6^ K^−1^ at 1500 °C and 2500 °C, respectively [15]. 

Density is an important factor for aerospace applications, as less weight is advantageous for flight; C/C has a lower density which depends on the volume fraction of fiber to matrix. ZrC has a density that is on the order of three times larger, but the density of ZrC is significantly less than other alternatives such as nickel-based superalloys [8,13].

We note also that the thermal conductivity in C/C is a directionally dependent property. Luo et al. measured thermal conductivities at room temperature between 7.5 and 21 W m^−1^ K^−1^ in the Z-direction and 31 to 47.5 W m^−1^ K^−1^ in the X-Y plane. The range changes to 3.5 to 22.5 W m^−1^ K^−1^ in the Z-direction and 27.5 to 50 W m^−1^ K^−1^ in the X-Y direction at 900 °C [20]. For ZrC, the thermal conductivity increases from 17.5 to 31 W m^−1^ K^−1^ at room temperature to 27.5 to 37.5 W m^−1^ K^−1^ at 1500 °C (the range of values corresponding to different experimental results) [21].

The experimentally reported value of flexure strength at room temperature is 140 ± 8 MPa and 460 ± 24 MPa for commercial C/C and ZrC, respectively. Data were not found for the flexure strength of C/C at higher temperatures, but the flexure strength of ZrC increases to 494 ± 44 MPa at 1600 °C before decreasing to 366 ± 46 MPa at 1800 °C; this unusual behavior is further discussed in a later section [11,22]. 

The effective elastic modulus of C/C ranges from 43 to 240 GPa by Windhorst and Naga et al.; the range reflects differences in processing conditions and volume fraction of fibers to matrix [13,23]. Comparatively, ZrC has a larger elastic modulus reported by Zhang as 434.9 GPa at room temperature and decreasing to 334.3 GPa at 1227 °C and 277.2 GPa at 1827 °C, which is higher than the upper bound reported for C/C [24].

The increase in mechanical strength of ZrC is advantageous for wing leading edge applications. At the wing tip, the pressure and temperature profiles become extreme in comparison to other components on the fuselage or body of the aircraft. Material properties degrade at high temperatures, so incorporating in a hypersonic vehicle ZrC that maintains its strength and stiffness at these temperatures is advantageous. 

### 1.2. ZrC Crystal Structure

Crystallographic stability across a wide temperature range is important for hypersonic applications. ZrC has an FCC rock-salt structure involving bonding between Zr-Zr and Zr-C atoms; no bonding between C-C atoms is reported. This is illustrated in Figure 3, where the bonding of electrons is primarily between the C-2p and Zr-4d energy states and is derived using Density Functional Theory (DFT), which is a computational quantum mechanical model used to predict material properties [4,25]. 

The rock-salt structure plays a pivotal role in determining the mechanical and thermal properties of ZrC—especially at high temperatures. ZrC demonstrates desirable mechanical properties for hypersonic applications. The high-temperature strength, hardness, and relatively low CTE are well suited for extreme environments. ZrC’s mechanical properties arise from the strong Zr-C covalent bond. The strength of this bond limits expansion at high temperatures as well as creates a robust material that is strong and resistant to wear. The second bond in ZrC—the metallic Zr-Zr bond—makes ZrC more thermally conductive than many other ceramics. The presence of a metal bonding allows a free flow of electrons throughout the metallic bonding. This allows ZrC to conduct through electrons as well as phonons. While most ceramics can only conduct mainly through the latter, ZrC’s ability to leverage both allows it to efficiently transfer heat more effectively than other ceramics. 

To provide insights into experimentally measured material properties in ZrC, computational models need to consider variations in the Zr:C ratio and the role of vacancies and defects that are readily incorporated into the material on account of carbon deficiency. Variations in the Zr:C ratio alter the materials’ properties. Depending on how this is leveraged, it can be detrimental or beneficial. For example, a carbon deficiency, the more common stoichiometric imbalance (Zr:C > 1), would introduce vacancies in the lattice. The addition of point defects leads to a reduction in thermal conductivity by electron and phonon scattering. This decrease in thermal conductivity will make the material more susceptible to thermal shock. As discussed in Section 3.2.1, the ratio of Zr:C alters the lattice parameter. This occurs due to asymmetrical bonding when there are carbon vacancies. A slight carbon deficiency increases the lattice parameter and, in turn, decreases the bond strength. A decrease in bond strength will be detrimental to mechanical properties. For all practical applications, it is ideal to have a 1:1 ratio. Further, these distorted lattice parameters influence phonon vibrations, and thus, the thermal conductivity, and slip dislocation motions, which could lead to unpredictable mechanical behavior. 

Furthermore, vacancies facilitate point defect diffusion. As shown by Yang et al., at elevated temperatures, Frenkel Pair defects distort the lattice. Defects in a material can result in premature failure caused by embrittlement or cracking. However, as a ceramic, ZrC exhibits brittle fracture mechanics and little plastic deformation before fracture. Due to this, the defect density in the material will determine its strength, as any defects can act as a point for stress concentrations and, ultimately, failure. Defects also limit thermal conductivity by contributing to electron scattering within the material. These can be reduced mainly via processing. Variations in sintering procedures or sintering aids can have large effects on ceramic materials. For example, if heating occurs too quickly during sintering, microcracking will occur. This ultimately will lead to stress concentration and early failure of the material. Compounding the mechanical concerns, if the interstitial species are reactive, these defects could result in degradation of the chemical stability and oxidation behavior of ZrC, negatively impacting the expected performance of ZrC as an aerospace material [26]. 

### 1.3. ZrC Sintering

The sintering of ZrC presents several challenges. ZrC has low self-diffusion, a high rate of grain growth at elevated temperatures, and an oxide layer covering the surface of the powders readily forms [27]. The low self-diffusion causes decreased densification, while the high rate of grain growth can yield porosity in the microstructure. Sintering of ZrC is mostly accomplished under high pressure to promote densification. Spark plasma sintering (SPS) or hot-pressing leads to the formation of highly densified ZrC. For SPS sintering, 65 MPa and 2100 °C are required when no sintering aids are employed. With sintering aids, 100 MPa and 1700 °C are sufficient for densification; the temperature reduction of 400 °C is advantageous from an industrial and commercial perspective [28]. The influence of pressure and high temperature on improving the physical properties of composite materials on account of high densification and nanosized grain boundaries has been reported by Tishkevich et al. for the case of W-Cu composites [29].

The pressureless SPS sintering of ZrC has been attained by using MoSi_2_ as a sintering aid. Sciti et al. studied the mechanical behavior of SPS ZrC containing MoSi_2_ in the volumetric range of 0–9 vol% at temperatures between 1750 and 2100 °C [28]. The addition of MoSi_2_ resulted in a decrease in the sintering temperature, refinement of the microstructure, and improvement of the mechanical properties in comparison with the monolithic material. MoSi_2_ forms a liquid phase along the grain boundaries during sintering, which is responsible for the temperature reduction necessary to densify ZrC. As the density is inversely proportional to particle size, for pressureless sintering, the average particle size must be decreased from microns to nanometers. This is achieved through mechanical and chemical processes as shown, using ball milling for both ZrC and ZrB_2_ [27,30]. With MoSi_2_ strengthening already fine grains, the mechanical properties are improved.

## 2. ZrC Material Properties of Interest for WLE

The ZrC mechanical, thermal, and chemical properties of interest for WLE applications are summarized in Table 2. Mechanisms that affect these properties are discussed, and potential solutions for improvement are proposed. 

### 2.1. Mechanical Properties

#### 2.1.1. Elastic Constants

Table 3 provides room temperature values of elastic constants for ZrC, derived from DFT and from experimental measurements. A local density approximation (LDA) that focuses on electronic density only is used to predict the values of *C*_11_, *C*_12_, and *C*_44_. With these values, Equations (3)–(5) can be used to calculate a shear (*G*), bulk (*B*), and elastic (*E*) moduli respectively. The shear modulus is determined by a Hill average, *G_H_*_,_ (Equation (3)), which considers a Ruess, *G_R_*, (Equation (2)) and Voigt, *G_V_*, (Equation (1)) average. This is analogous to considering an iso-stress vs iso-strain average then averaging the two cases to determine a shear modulus value [31,32,33,34].
(1)$$G_{V}=\frac{C_{11}-C_{12}+3C_{44}}{5}$$
(2)$$G_{R}={\left[\frac{4C_{44}+\left(C_{11}-C_{12}\right)\;}{5C_{44}\left(C_{11}-C_{12}\right)}\right]}^{-1}$$
(3)$$G_{H}=\frac{G_{V}+G_{R}}{2}$$
(4)$$B=\frac{C_{11}+2C_{12}}{\;3}$$
(5)$$E=\frac{9BG_{H}}{\;3B+G_{H}}$$

The experimental values compare well with DFT calculations. Based on the elastic constant values, the elastic modulus for ZrC is around 400 GPa. The elastic properties become important design features as a high Young’s Modulus correlates with high stiffness. It is expected that a ceramic material experiences very little plastic strain and therefore exhibits essentially no plastic deformation prior to failure. Further, ceramics have fracture toughness inferior to metals. A modulus that is too high could result in premature fracture, and a modulus that is too low leads to facile deformation. 

#### 2.1.2. Flexure Strength

For WLE applications, material strength is an important property. Andrievski et al. [37] reported ZrC flexure strength from room temperature through 2500 °C. Their results show that ZrC has a peak flexure strength at 1800 °C and reduces to room temperature strength values above 2500 °C. Demirskyi et al. [11] demonstrated that the ZrC flexure strength can be increased by a factor of two. They synthesized utilizing SPS, a single-phase (Ta,Zr,Nb)C ceramic (TZN-3), fabricated from TaC, ZrC, and NbC precursors at 1920 °C, and the flexural strength was measured at temperatures from 23 °C to 1800 °C. They found that the flexure strength increased up to 1600 °C; at higher temperatures, it decreased to values lower than for monolithic ZrC. Demirskyi suggested several explanations for the differences in peak strength between ZrC and TZN-3, noting that the porosity was nearly 5% in the TZN-3 sample; the elastic modulus observed in the loading curves was markedly lower; and, the fracture behavior above 1600 °C was predominately transgranular as opposed to mixed fracture behavior below. Figure 4, compiled by Demirskyi et al. [11], compares flexure strength vs. temperature for ZrC and TZN-3. Table 4 provides representative values of key mechanical properties for these ceramic materials.

The increase in strength at elevated temperatures for both TZN-3 and ZrC is attributed to augmented microplasticity that occurs as stresses are relaxed due to the temperature increase. However, the rapid decrement at above 1600 °C in TZN-3 is attributed to solid-solution alloying and grain size growth [37]. A similar decrease was reported for ZrC between 2000 °C and 2500 °C. While no explanation was provided, Harrison and Lee reported a sudden reduction in Young’s Modulus in this temperature range [4]. Furthermore, given the high temperatures and fine grains of the initial ZrC measurements, it is possible that grain growth occurred during testing, which may explain the reduced flexure strength for the 2500 °C measurement. Given the context of hypersonic applications, a decrease in fracture strength would be undesirable at ultra-high temperatures, but the decrease in ZrC at 2500 °C is quite comparable to values measured at room temperature with error that is on the order of five times larger than the room temperature data. Also, materials may experience weakening at ultra-high temperatures caused by elongation of bonds and lattice parameters associated with CTE increases at high temperatures. 

It is noted that fully dense materials have higher strengths since failure in a bulk material is nucleated at defects such as pores [11]. Thus, to enhance the strength of ZrC, the use of sintering aids or the synthesis of multicomponent ceramic material have been shown to be successful in increasing its strength.

#### 2.1.3. Fracture Toughness

Due to their brittle nature, fracture toughness is an important property to consider when using ceramics. A high fracture toughness is desired since ceramics have almost no ability to slow crack propagation through plastic deformation. 

While an increase in flexure strength has been observed at elevated temperatures, a decrease in fracture toughness is observed due to a lack of ductility of ZrC at these high temperatures. Zhang et al. [24] modeled the fracture toughness behavior vs. temperature based on Niu et al. [38] and Mazhnik et al. [39]. The model predicts fracture toughness to decrease over the 0–2250 °C temperature range, as shown in Figure 5. 

One potential method to increase the fracture toughness of ZrC is to make a C/C–ZrC composite. The fiber of the C/C composite improves the fracture toughness of the ZrC due to its inability to plastically deform. Adding fiber reinforcement can aid in limiting catastrophic failure and reduce the material density. As cracks form and spread, the fibers can bridge cracked regions, counteracting the formation of larger cracks; however, C/C would be exposed, which is less chemically stable than ZrC in aggressive oxidizing environments. The important trade-off for aerospace applications is the prevention of catastrophic failure and to lower material density vs. oxidation and corrosion resistance. As described by Wang et al. [16], C/C–ZrC composites fabricated by reactive melt infiltration improve fracture toughness as well as ablation resistance in oxidizing environments and decrease the density of ZrC. The lower density and ablation resistance are attractive improvements for hypersonic applications. The process to make C/C–ZrC composites uses a carbon fiber felt that is first densified by carbon to obtain a porous C/C skeleton, which is then infiltrated with a Zr melt at temperatures higher than the melting point of Zr. The zirconium melts into the porous C/C preform and reacts with excess carbon to form ZrC. The process must be optimized to ensure the correct Zr-to-C ratio, since, as previously discussed, material properties could be different if the composite was carbon-rich. 

### 2.2. Thermal Properties

#### 2.2.1. ZrC Melting Point

The friction caused by air resistance at hypersonic speeds heats up vehicles to extremely high temperatures. NASA reports that the space shuttle traveled at hypersonic speeds during its re-entry into the atmosphere, exceeding 17,500 miles per hour [40]. The air friction at this speed causes the leading edge of the shuttle to experience ultra-high temperatures beyond 2000 °C. With a melting point of 3427 °C, ZrC exceeds design requirements for a WLE in hypersonic applications [1,14,41]. While applications of ZrC for WLE in hypersonic flights is the focus of this review, given the chemical stability and thermal–mechanical properties, ZrC is a very attractive material for applications in heat exchangers and thermal protection systems (such as ablative skins). Figure 6 shows nominal surface temperature profiles experienced on other sections of the Space Shuttle Orbiter [14]. The temperature profile shown in the figure is merely representative, and temperature profiles of varying re-entry vehicle geometries may experience higher temperatures based on speeds and fundamental aerodynamics. 

The strength of the atomic bonding in ZrC accounts for its high melting point. It exhibits metallic bonding between zirconium atoms and covalent bonding between carbon and zirconium atoms. The covalent bond has a bond energy of 561 kJ mol^−1^. In comparison, SiC—another industry standard high-temperature material—exhibits a bond energy of 444 kJ mol^−1^ between silicon and carbon atoms. The strong covalent bonding in ZrC is responsible for its high melting temperature [3]. 

#### 2.2.2. Coefficient of Thermal Expansion (CTE)

At ultra-high temperatures, the coefficient of thermal expansion (CTE) is a key material property for WLE applications. Dimensional stability is critical to design safe and reliable components made from ZrC. The CTE vs. temperature of ZrC is presented in Figure 7 [15]. 

As the material heats up, it expands and undergoes a volumetric change, which ultimately changes the material density, *ρ*—one of the parameters that determine the thermal conductivity, κ, which is given by the thermal conductivity equation, *κ = α . ρ . C_p_* (*α* is the thermal diffusivity, and *C_p_*, the specific heat) [17]. The equation indicates that a reduction in density decreases the thermal conductivity. CTE and thermal conductivity, in addition to fracture toughness and elastic modulus, provides an indication of a material’s ability to resist thermal shock. If heat can diffuse away rapidly prior to contraction/expansion, or if the material is stiff enough to resist the load, then the material can better tolerate thermal shock [42,43]. 

#### 2.2.3. ZrC Dimensional Stability at High Temperatures

CTE mismatch is one of the most important criteria in the selection of coatings for oxidation and corrosion prevention, as a large mismatch between the ceramic material and the overcoat will result in cracking and delamination. Creating layered or graded coating structures is a useful way to alleviate the CTE mismatch. Interlayers of a material having an intermediate CTE between that of the coating and substrate can be intercalated between the two in order to reduce the CTE mismatch and relieve stress. However, this is not always possible in hypersonic applications. Due to the hypersonic operating environment, any material used in a coating will have to be mechanically and thermally robust, thus limiting the number of materials needed for a coating is generally preferred to optimize interface bonding. Some silicides, such as MoSi_2_ as demonstrated by Zhang, Sun, and Fu, can be processed to form a layered coating. The first layer, Mo_3_Si_5_, has a CTE coefficient between the substrate and the coating material MoSi_2_. This is beneficial because now there is only one bonding interface as well as a CTE gradient [44].

Dimensional stability and CTE matching between ZrC and protective overcoats are important requirements for high-temperature applications, to prevent oxidation and ensure chemical stability. Expansion and contraction from heating and cooling can induce micro-cracking between the coating and the UHTC material, ultimately resulting in delamination and catastrophic failure. Micro-cracks can act as short-circuit pathways for oxygen to diffuse. Ultimately, the formation of cracks and delamination severely compromise the required material properties for UHT environments [45]. 

To prevent cracking at interfaces of different materials, careful consideration to material selection is given to best match CTEs. Commonly, SiC coatings are used to protect against oxidation in high-temperature environments’ materials used in energy and aerospace applications. The CTEs of several ultra-high temperature materials are shown in Figure 8. WC and SiC have lower CTE values than Y_2_O_3_, ZrC, and MoSi_2_. The CTE compatibility of different materials employed in composites or coatings is critical as large differences in CTE result in micro-cracking and failure at the interface, compromising oxidation and corrosion resistance, and mechanical integrity [15,46]. 

#### 2.2.4. Thermal Conductivity 

The thermal conductivity vs. temperature of ZrC is shown in Figure 9 [4]. It is ~20 W m^−1^ K^−1^ at room temperature, and it increases at higher temperatures. This is counterintuitive as, in ceramic materials, the thermal conductivity typically decreases at higher temperatures. Harrison and Lee [4] proposed mechanisms for the thermal conductivity in ceramics and ascribed two processes for heat transfer: electron and phonon conduction. Since most ceramics exhibit no metallic bonding, heat is transported by phonon conduction. At high temperatures, vibrations within the lattice are so strong that phonon transport collapses, causing a decrease in thermal conductivity. ZrC exhibits the opposite behavior, as observed in Figure 9. 

As phonon conduction contribution is unimportant at high temperatures, the conduction band electrons must be responsible for heat transfer—indicative of metallic bonding at these high temperatures. Metallic bonding allows for delocalized electrons to move easily throughout the lattice. Since electrons can move freely, they can transfer heat more effectively than electrons in covalent or ionic bonds. As Harrison and Lee illustrate in Figure 9, there is a high degree of variability between measurements in different studies. The variability can be attributed to materials processing differences. Samples with higher densities—less porosity—will exhibit higher thermal conductivity due to less phonon scattering, therefore, Harrison and Lee corrected the data to reflect 100% theoretical density for each specimen by using the Maxwell-Eucken Equation in Equation (6) to determine to determine thermal conductivity of a 100% dense sample, where *P* is the porosity, *K_p_* is the measured thermal conductivity value, and *K_TD_* is thermal conductivity adjusted to full-density. Since this equation is an approximation, fluctuations in thermal conductivity in the original samples could account for the variation [4].
(6)$$K_{p}=\frac{\left(1-P\right)}{\left(1+P\right)\;}K_{TD}$$

Ceramic materials transport thermal energy through phonon conduction, which are ordered lattice vibrations. At high temperatures, these phonons vibrate so quickly that the phonons become unstable and begin to scatter. Since phonons are no longer highly ordered, they no longer conduct heat effectively. Due to this, most ceramics show a decrease in thermal conductivity at high temperatures. However, Figure 9 shows that the opposite is the case for ZrC. Even above 2000 K, the thermal conductivity is still increasing. This can be attributed to the metallic bonding present between Zr atoms in the material. While ceramics are dominated by phonon conduction, metallic bonding allows for conduction via electrons. As temperature increases, the overall increase in energy allows electrons to move much easier throughout the material. This increase in electron conductivity in turn increases the thermal conductivity in the material.

### 2.3. Chemical Properties

#### 2.3.1. Thermodynamic Stability of ZrC

The chemical stability of ZrC is important for high-temperature environments. For hypersonic applications, oxidation is the factor that most affects the chemical stability of ZrC at high temperatures. The thermodynamic affinity for oxygen increases with temperature, creating issues for ultra-high temperature applications [47]. Applying a thermodynamic analysis, the Gibbs energy for reactions between zirconium and several common elements such as carbon, boron, nitrogen, and oxygen can be estimated as a function of temperature. As the ZrC system is deficient in carbon, the reaction at ultra-high temperatures can be predicted using an analytic thermodynamic calculation with the reaction equation for zirconium and a nonmetallic element (C, B, N_2_, or O_2_).

Reactions of with C, B, N_2_, and O_2_ and reactions products:
Zr +C →ZrC$Zr+2∙B\to ZrB_{2}$$2∙Zr+N_{2}\to 2∙ZrN$$Zr+O_{2}\to ZrO_{2}$


The Gibbs energy of reaction can be solved in a standard, reference state for this reaction, where the reference state is the phase of the material at a given temperature under standard atmospheric conditions. The Gibbs energy of formation can be obtained from various databases such as the Thermophysical Properties of Materials Database compiled by Barin, and the Gibbs energy of reaction can be calculated [48]. Since WLE applications require a wide ultra-high temperature range, compounds that exist as solids above 2000 °C were considered. Figure 10 shows the results of the analysis.

For the temperature range considered in Figure 10, ZrO_2_ is thermodynamically the most favorable reaction, and ZrC is the least favorable one. This indicates that oxidation protection of ZrC requires protective coatings, alloying, or doping to stabilize ZrC and prevent the formation of ZrN, ZrB_2_, or ZrO_2_, which would compromise the ZrC physical properties. Additionally, if ZrC has a carbon-deficient stoichiometry, this analysis suggests that excess Zr is more likely to bond with another species than remain free in the ceramic, which would reduce the oxidation resistance of the system and create more defects, which would change the mechanical and thermal properties of the system. By forming compounds with free zirconium, for the case of a carbon-deficient (ZrC_1−x_) system, via alloying or doping, the reaction energy between zirconium and corrosion species in the atmosphere would be reduced. Thus, the efficacy of a corrosion-protective coating would be enhanced provided that minimal cracking occurs.

#### 2.3.2. Reaction Kinetics

For the environments in which hypersonic vehicles travel, oxidation kinetics and chemical stability are critical for reliability and safety. An example of the severity of oxidation of ZrC at elevated temperatures is shown in Figure 11. Gasparrini et al. [18] oxidized a sample of ZrC heated to 1000 °C for 30 min in air, and an oxide layer approximately 0.25 cm thick formed. This demonstrates the need for protective coatings to prevent ZrC oxidation at elevated temperatures. The choice of protective coating needs to be carefully considered as it is intended to prevent oxidation and ablation; however, adding additional material adds mass to the vehicle, which could hinder fuel economy and performance.

From the work of Gasparrini et al., ZrC exhibits linear oxidation kinetics, as shown in Figure 12 [18]. This indicates that chemical reaction is the rate-limiting step; hence, the diffusion of zirconium (or carbon) to the surface is unlikely to be the oxidation reaction driver. Knowledge of this rate-limiting step suggests that a coating to prevent the surface of ZrC from interacting with air can be effective in hindering oxidation. 

It is noted that in comparison with C/C composites, ZrC exhibits better oxidation resistance up to 600 °C, where ZrC remains chemically stable, whereas C/C composites readily oxidize above 500 °C [19,49]. The work by Goto et al. [18] indicates that the oxidation of C/C composites follows a parabolic rate law, as shown in Figure 13, and that the oxidation is controlled by the transport of gaseous species to and away from the surface of the C/C composite, resulting in mass loss (note that oxidation of C/C composites is measured as weight loss, whereas for ZrC, it is measured by oxide thickness). The effect of oxidation of C/C is more troublesome than that of ZrC as gaseous diffusion will continuously transport species to the surface; whereas for ZrC, the mechanism requires the transport of species through an oxide layer, which hinders growth kinetics. 

## 3. Discussion

ZrC offers some attractive physical properties for hypersonic applications; however, improvements on other properties are needed to potentially replace C/C materials. Some of these improvements can be implemented in the synthesis and/or processing of ZrC, while other properties could be engineered in postprocessing steps. The microstructure of ZrC plays a key role in its physical and chemical properties. For example, synthesis of small particles is expected to increase mechanical strength; however, a concomitant increment in grain boundaries occurs, and they can become conduits for diffusion of oxygen or other reactive species. This implies that engineering improvements for chemical, mechanical, and thermal properties need to consider their impact on overall physical properties. Sintering is employed to densify ceramics as pore elimination improves mechanical and thermal properties; however, the incorporation of sintering elements into the ZrC lattice as defects and at grain boundaries could also compromise the chemical stability of the compound. Alloying ZrC with other ceramic materials and forming composites offers pathways for physical property improvements. This section reviews research in ceramic materials that can be explored to improve ZrC physiochemical properties.

### 3.1. Mechanical 

#### 3.1.1. High-Entropy Ceramics

High-entropy oxides maximize the configurational entropy to stabilize equimolar element mixtures and achieve more robust systems. Entropy stabilization of multicomponent oxide materials was first demonstrated in 2015 and was rapidly followed by other high-entropy disordered ceramics and proven to be useful as thermal barrier coatings, wear-resistant and corrosion-resistant coatings [50]. High-Entropy Ceramics (HEC) potentially address material deficiencies by incorporating into a single-phase structure elements with a wide range of properties that collectively improve a diversity of material properties. Figure 14 shows an Ashby Plot for thermal conductivity vs. elastic modulus of HECs compared to other materials [51]. 

To increase the strength of ZrC, sintering aids are employed as they enhance the relative-density, high-entropy ceramics with multiple components, offering an additional route, as indicated in the figure. Alloying leads to an increase in strength due to induced strain in the crystal lattice, but this comes at the expense of ductility. The increased lattice defects may strengthen the material, but they serve to embrittle it too. Demirskyi et al. synthesized a single-phase, high-entropy ceramic (HEC) carbide with tantalum, zirconium, and niobium in equal parts. This ceramic had room temperature strengths of 460 ± 24 MPa, and exhibited a strength of 366 ± 46 MPa at 1800 °C [11]. 

Castle et al. synthesized two high-entropy ultra-high temperature carbides using zirconium: (Hf-Ta-Zr-Ti)C and (Hf-Ta-Zr-Nb)C, formed through SPS. These HEC were highly dense with high purity and formed into a single-phase compound. They demonstrated higher hardness, 36.1 ± 1.6 GPa, compared to monolithic HfC, 31.5 ± 1.3 GPa, and ZrC, 31.3 ± 1.4 GPa, or the binary (Hf-Ta)C ceramic, 32.9 ± 1.8 GPa. Undoubtedly, HEC presents a new paradigm for the discovery of materials offering superior properties in high-temperature aggressive environments [52]. 

For hypersonic applications, a specific area of research that needs to be addressed is the single-phase stability of HEC across the wide temperature range experienced by ceramic components in a hypersonic vehicle. The single-phase HEC is formed at high temperatures that maximize the entropy; the multicomponent material is then rapidly quenched to stabilize the single phase.

Exposing the ceramic to high temperatures can reverse the single-phase formation and yield a multiphase material under slow cooling-down conditions. However, diffusion in these materials is strongly hindered by the multiple elements present. To quantify their thermal stability, kinetic studies under hypersonic vehicle temperature excursions need to be conducted [52]. 

#### 3.1.2. Sintering and Densification Improvement

Sintering aids facilitate densification. Using MoSi_2_ has been reported to improve the mechanical properties of ZrC processed with SPS. When increasing amounts of MoSi_2_ (1, 3, 9 vol %) were added to ZrC, the relative density increased, the grain size decreased, and the overall mechanical properties, namely, strength and hardness, increased, as shown in Table 5 [28]. 

Of note is the decrease of the elastic modulus with additions of 1% and 3% of MoSi_2_; however, at 9%, the modulus is the same as that of pure ZrC. The flexure strength, on the other hand, increases with additions of MoSi_2_. This change was attributed to internal stresses induced by the CTE mismatch between ZrC and MoSi_2_, 6.7 × 10^−6^ K^−1^ for ZrC and 8.9 × 10^−6^ K^−1^ for MoSi_2_. Both the microstructure changes and the increase in toughness ultimately increased the flexural strength of the material as MoSi_2_ volume increased [28]. 

A similar effect was observed by Zhao et al. [26] in pressureless sintering of ZrC with carbon and silicon used as sintering aids to form a ZrC-SiC alloy. Four ZrC starting powders were employed: (1) As-received ZrC, (2) ball-milled ZrC (MZ), (3) ZrC + 2 wt % graphite (ZC), and (4) ZrC + 20 vol % SiC (ZS). The properties and densities are provided in Table 6. 

The data in the table indicate the benefit of ball milling, which reduces the average powder grain size, thereby increasing the relative density. The addition of carbon at 1900 °C reduces the oxides present to prevent impurities from hindering densification [27]. 

#### 3.1.3. Zirconium-Carbide-Based Composites

A significant amount of research has been conducted on carbon-fiber-reinforced ZrC composites. Chen et al. [53] synthesized C/C–ZrC–ZrB_2_ composites through slurry infiltration (SI), precursor infiltration and pyrolysis (PIP), and reactive melt infiltration (RMI). While unhomogenized zirconium particles are an issue in noncomposite ZrC (reducing mechanical properties) in ZrC composites, free zirconium aids in increasing the material density and, thus, its fracture toughness. C/C–ZrC–ZrB_2_ composites exhibited higher density than C/C composites, 3.07 g/cm^3^ vs. 1.4 g/cm^3^. Further, mechanical testing was performed on the C/C–ZrC composites, as shown in Figure 15. 

The flexure strength and elastic modulus of 147 MPa and 27.7 GPa, respectively, are reported [53]. This is only about 25% of the value reported for monolithic ZrC sintered through SPS [28]. While lower values were obtained for these two properties, the composite exhibited ductile fracture, shown in the micrographs in Figure 16. A ductile fracture requires more energy for the crack to propagate through and should allow any cracks to be identified prior to catastrophic failure at the WLE [53]. The added crack deflection of the C/C–ZrC composites represents a key synergy of C/C and ZrC with a lower density compared with ZrC. Without the C/C fibers, the ZrC ceramic would have failed in a brittle manner without any crack deflection to blunt crack growth. This tends to present a situation where cataphoric failure occurs.

The C/C composite fibers elongated during stress testing, resulting in greater material ductility. In the images of Figure 16, fiber pullout is identified, and bending of the fibers can be noticed. This behavior is not typical of a ceramic, and, depending on the application, it provides additional engineering design parameters [53]. 

### 3.2. Thermal Properties

#### 3.2.1. Melting Point Increase by Lattice Parameter Reduction

ZrC has a melting point that is suitable for most extreme applications such as hypersonic flight. The melting point can potentially be further increased by decreasing the Zr-C lattice parameter. The C/Zr ratio, as shown in Figure 17, modifies lattice parameters. For a C/Zr ratio of 1 and greater, the lattice parameter is less than 4.7 Å. Just below a ratio of 1.0 (C = 45%), the lattice parameter decreases, and the ZrC melting point is at a maximum. At lower C/Zr ratios, the lattice parameter increases, and the melting point subsequently decreases. As the lattice parameter decreases, the bonds become stronger, yielding a higher melting point [3]. 

Another approach to decrease the ZrC lattice parameter is through doping it with oxygen or boron. Work by Harrison and Lee [4] demonstrated that doping these atoms into ZrC results in fewer lattice vacancies than in stoichiometric ZrC. The atomic radius of oxygen is 14.3% smaller than carbon, hence interstitial substitution results in shorter bond lengths, incrementing covalent bond strength and therefore a higher melting temperature. Moreover, in Section 2.3.1, both ZrB_2_ and ZrO_2_ were demonstrated to be more thermodynamically stable than ZrC. If excess zirconium is bonded to boron or oxygen, the high-temperature behavior might be more stable and predictable. 

#### 3.2.2. Doping to Improve Thermal Conductivity

Thermal conductivity is associated with carbon vacancy density in the ZrC lattice. At high temperatures, thermal conductivity of ZrC is driven by conduction-band electrons from metallic bonding. Since these vacancies scatter electrons, they hinder thermal conduction [31]. This can be circumvented by doping. Huang et al. found that the effect of carbon vacancies can be mitigated by doping ZrC with oxygen or boron [54]. These dopants increase thermal conductivity in two ways. First, by filling the carbon vacancies, fewer electrons are scattered. With fewer electrons being scattered, heat can be conducted more efficiently. The study also found that these dopants can also increase the stability of phonon conduction. Doped ZrC was modeled for different dopant configurations through quasi-harmonic approximation. As the vacancy density increases, the frequency in the acoustic phonon branch decreases, which lowers thermal conductivity. Filling the vacancies with boron or oxygen increases the acoustic phonon frequency, mitigating the effect of vacancies. Thus, ZrC_0.75_O_0.25_ and ZrC_0.75_B_0.25_ had higher thermal conductivities than ZrC_0.75_, the carbon-deficient model. However, these doped models did not exhibit higher thermal conductivities than ZrC with a C/Zr ratio of 1 [54].

### 3.3. Chemical Properties

#### 3.3.1. Chemical Stability

ZrC is prone to oxidation at high temperatures, therefore, a barrier coating will be needed in harsh oxidizing environments. To prevent issues such as micro-cracking at the interface between ZrC and the coating due to expansion and contraction under the broad temperature range experienced by the hypersonic vehicle, a dopant that does not negatively impact the coefficient of thermal expansion of ZrC needs also to be considered. In Figure 18, ZrC doped with boron and oxygen is shown to increase the temperature dependence of CTE and the change in volume of the material. ZrC_0.75_ shows the lowest temperature dependence for CTE and volumetric change. This could be due to its relatively low melting point compared to the other compositions. Due to the relatively high density of C vacancies, bonding is weaker, leading to a lower melting point [54]. To minimize the effect of volumetric changes experienced by ZrC during intermediate temperatures of hypersonic flights, the effect of dopants and coatings that have similar magnitude CTE values as ZrC should be explored to minimize delamination. It is also plausible that some dopants upon reacting with the atmospheric species may form protective coatings.

TaSi_2_ can be considered as an alternative additive. It decreases the mobility of oxygen and other dislocation defects by reinforcing the grain boundaries with a tantalum solid solution. This is due to the high solubility of tantalum in zirconium. The excess carbon from ZrC forms bonds with the silicon in solution to form SiC, which mainly agglomerates at the triple points of the grain boundaries. As the silicon in solution mostly bonds with O_2_ in the sintering phase and then permeates out of the solution as SiO_2_, the result is increased densification. In oxidation tests, Silvestroni et al. [55] showed that ZrC samples demonstrated decreased oxidation up to 1800 K for the first 10 min by quantifying the concentration of SiO and CO_2_ produced. At 1800 K, the measured concentration remained below 0.1, while at 2000 K, the concentration was more than double, and at 2200 K, it was seven times larger. This is indicative of increased stability compared to monolithic ZrC as the additives retarded diffusion of gaseous species such as oxygen to the unreacted ZrC. 

#### 3.3.2. Oxidation Resistant Coatings

ZrC oxidizes at high temperatures but to a lesser extent than uncoated C/C. ZrC requires protective coatings as C/C composites, which typically use SiC coatings. Using SiC as a protective overcoat for ZrC is inadequate based on the coating’s melting point, strength, and propensity to crack [56]. Furthermore, if the overcoat CTE does not match well with that of ZrC, then short-circuit diffusion through cracks will accelerate oxidation [57,58]. Figure 19 depicts an example of CTE mismatch in a weakly bonded C/C with SiC coating. In Figure 19a, the coating around the fiber is cracked, exposing the fiber to the environment. In Figure 19b, the bond between the carbon fiber and SiC matrix is weak and produced a gap. This gap can allow for corrosion or oxidation to occur, which can compromise the required material properties for ultra-high temperature environments [45]. 

There are several coatings that can be employed to retard oxidation in ZrC at high temperatures that are being explored: ZrB_2_-SiC and SiC-ZrC-SiC coatings. ZrB_2_-SiC has shown oxidation-protection advantages over a simple SiC-coating [43,46,56]. 

Figure 20 shows improved oxidation resistance through the addition of ZrB_2_ to SiC coating at 1500 °C. While holding the coated graphite specimen at 1900 °C, the weight loss was greater for the sample coated with only SiC. With the addition of approximately 20% ZrB_2_, this weight loss is slowed down. Further, it is demonstrated that the coating is protective during thermal cycling at 1500 °C [56]. However, Mei et al. [45] reported that, while oxidation performance greatly improves, there is a surface interaction that produces a glassy phase, which is undesirable as it reduces the materials’ ductility and makes the coating brittle and prone to cracks. In addition, the presence of this glassy phase will render CTE mismatch issues harder to circumvent. 

SiC-ZrC-SiC multilayer coatings on C/SiC composites were studied as ultra-high temperature coatings. This is attractive for ZrC, as it reduces the CTE mismatch between the coating and substrate, which is commonly one of the factors leading to crack formation, delamination, and catastrophic failure. A fracture surface of a C/SiC composite is shown in Figure 21 with a SiC-ZrC-SiC applied coating. In this case, the ZrC deflected and arrested the crack, thereby preserving the physical integrity of the passivating exterior layer of SiC. Further, when exposed to oxidizing environments, the interior substrate was protected while the exterior served a sacrificial means [46]. 

Cracks are observed in the figure, which create fast diffusion pathways, accelerating oxidation. The authors do suggest that the inclusion of the ZrC layer promotes crack deflection which improves fracture toughness [46].

#### 3.3.3. Ablation

Surface ablation is employed to overcome hypersonic temperature profiles. Ablation is the primary method used by NASA for re-entry vehicles such as the Space Shuttle Orbiter; however, when a material is ablated, it is sacrificed and needs to be reapplied to the vehicle. Previous studies have investigated ZrC as an ablation protective coating on C/C composites which showed improved mass ablation rates compared to uncoated C/C samples [59]. 

Ablation coatings are applied to protect the substrate exploiting chemical reactions of the coating; dense oxide coatings can improve ablative properties. In a study, Luo et al. [60] discusses the need for more stable coatings above 1800 °C. Two coatings are considered for C/SiC–ZrC composites: Y_2_O_3_ and La_2_O_3_. It was experimentally determined that La_2_O_3_ provided enhanced ablative protection by forming a dense La_2_Zr_2_O_7_ layer that allowed for additional ablation cycles, whereas Y_2_O_3_ remained stable (i.e., did not form a compound or additional oxide layer), which resulted in no substantial improvement in the ablative performance. Depicted in Figure 22 is a schematic of the ablation mechanism for each coating. 

The schematic shows the formation of ZrO_2_ and LaZr_2_O_7_ during the ablation test. It is unclear what the effect on mechanical performance would be for both additional phases and a porous ZrO_2_ region. This study specifically looked at thermal properties and thermal protection systems [60]. 

## 4. Summary and Future Perspectives

### 4.1. Summary

Zirconium carbide shows great promise for hypersonic and aerospace applications. In this work, we evaluated ZrC as a candidate material for the WLE of hypersonic vehicles to replace C/C. A review of mechanical, thermal, and chemical properties of ZrC was provided. The physical properties most important for WLE applications are: flexure strength, fracture toughness, melting point, coefficient of thermal expansion, thermal conductivity, and oxidation properties.

Zirconium carbide has an ultra-high melting point, high fracture toughness, and stable strength at high temperatures. Specifically, in comparison to C/C composites, ZrC exhibits increased compressive strength and elastic modulus. Further, ZrC offers a better thermal match with potential protective coatings, and without coatings it is less prone to oxidation than C/C. Additionally, ZrC has better thermal conductivity due to metallic bonding between zirconium atoms. These properties are important in hypersonic applications. 

The physical properties of ZrC affect its functional properties in multiple ways: strong covalent bonding between carbon and zirconium atoms results in its high melting point and higher elastic modulus. Carbon deficiency in the compound not only affects mechanical and thermal properties but can allow defects and impurities to bond with zirconium atoms, modifying its chemical and physical properties. 

Therefore, optimizing ZrC for UHT applications requires a balanced approach not to negatively impact key attributes. Improving the properties of ZrC can be realized through alloying, forming composites, using sintering aids, or doping. However, improving one property may have a deleterious effect on other attributes, and finding a balance becomes an engineering challenge. Table 7 highlights optimization methods to improve specific properties as well as possible disadvantages inherent to such improvement method. 

### 4.2. Recommendations for Further Improvements of ZrC for WLE

C/C-ZrC composites are one of the leading solutions advocated in this work for WLE components and applications. There are many advantages to this composite system. The addition of C/C lowers the overall material density compared to monolithic ZrC, which is beneficial for aerospace applications as it reduces weight. Reduced weight translates to better fuel economy, lower carbon emissions, and better flight performance. Also, the inclusion of C/C promotes nonbrittle fracture and/or increases the ductility of the material. Fracture toughness is typically lower in monolithic ceramics compared with metals or composites as cracks have no resistance. In a C/C-ZrC composite, crack propagation is hindered by C/C fibers, thereby increasing the fracture toughness resulting in a change of the failure mechanisms from brittle to ductile.

### 4.3. Future Work

The next major challenge for hypersonic materials will focus on maintaining material stability. During mission execution, materials must travel through uncontrolled environments. It is difficult to understand all of the conditions a material will be exposed to, but it is possible to address some of the most extreme cases. Herein lies the next two major challenges: corrosion control and volumetric changes. Controlling the rate of corrosion and oxidation will increase the reliability of the material performance. As discussed in this work, with a non-stoichiometric compound, the chances of undesired reactions increase, so future work should be focused on methods to reduce the likelihood of interactions through doping, alloying, or coating ZrC. Likewise, ZrC has a known volumetric change resulting from a change in crystal structure during intermediate temperatures of hypersonic flight. These volumetric changes impact the bonding in a composite or with a coating. Ultimately, finding a solution that minimizes the volume change will allow for ZrC to be more widely used in aerospace components and will improve the geometric compatibility with other parts. 

### 4.4. Conclusions

This review investigated the current state of the art of ZrC for WLE applications in hypersonic flights. The review covered mechanical, thermal, and chemical properties, while comparing ZrC to C/C. Several techniques were introduced to augment or to engineer its properties to improve various characteristics and requirements for hypersonic flight applications. Outside of hypersonic applications, ZrC is an attractive candidate for nuclear reactor fuel rod cladding. The functional requirements of cladding materials are not dissimilar from hypersonics. They must have very high melting points as well as strength and be able to retain these properties at high temperatures. It also requires the ability to resist thermal shock, a property of ZrC which is superior to other similar ceramics due to its high-temperature thermal conductivity. Most importantly, the CTE between the cladding and the fuel material is housed within.

## Figures and Tables

**Figure 1 materials-16-06158-f001:**
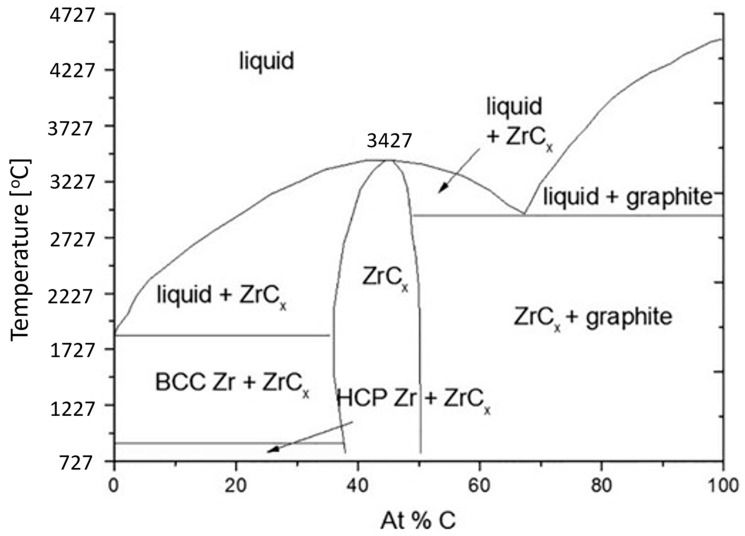
Zr-C phase diagram [4].

**Figure 2 materials-16-06158-f002:**
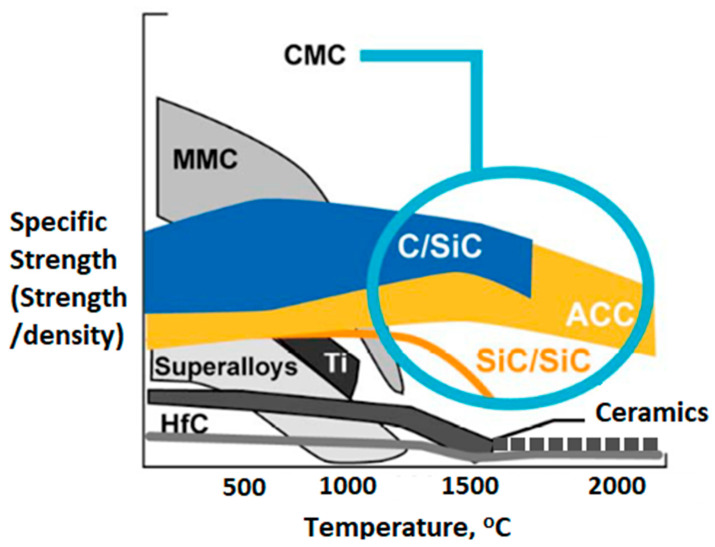
The specific strengths (SE) of various WLE candidate materials at elevated temperatures. Metal Matrix Composites (MMC) have initially high SE, but it rapidly decreases with increasing temperature. Ceramic Matrix Composites (CMC) tend to have stable SE through a wide temperature range. In super alloys such as MMCs and other metals, SE decrease rapidly near the melting point. Advanced C/C Composites (ACC) and C/SiC composites generally exhibit high melting points and low densities and maintain high SE over a wide temperature range. While having a generally lower SE, ceramics can potentially exhibit constant strength. There are some ceramics of interest above 1500 °C, such as ZrC, that have comparable melting points to ACC materials and are mechanically stable [14].

**Figure 3 materials-16-06158-f003:**
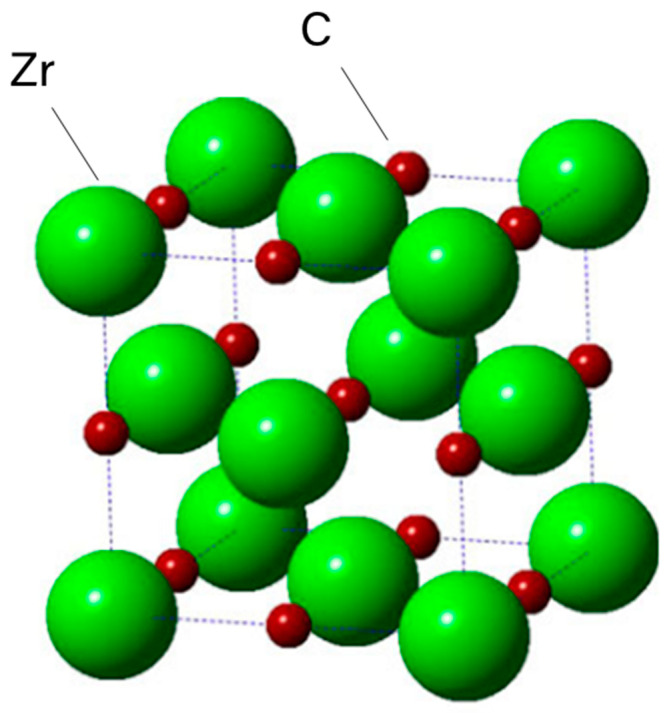
Density Functional Theory Model of ZrC showing a rock salt bonding lacking C-C bonding [4].

**Figure 4 materials-16-06158-f004:**
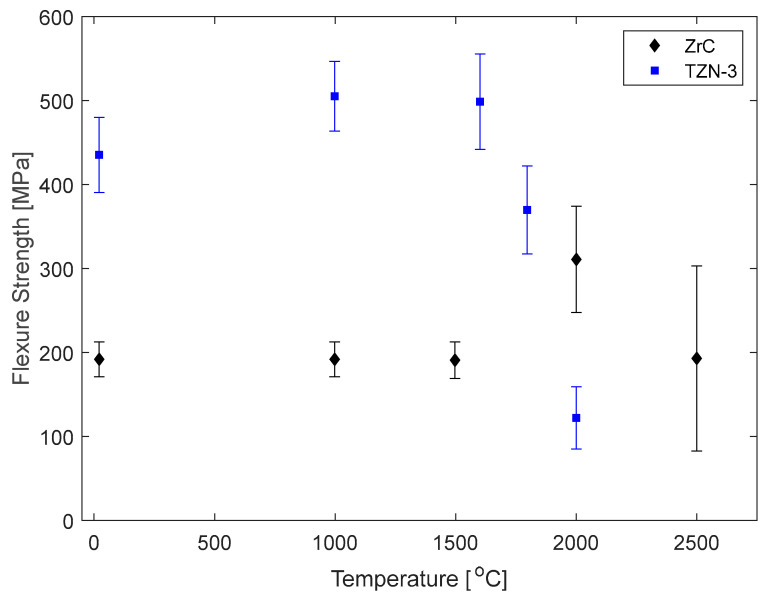
Flexure strength vs. temperature (°C) for monolithic ZrC (diamonds) and TaC, ZrC, and NbC alloy—TZN-3 (squares)—adapted from Demirskyi [10]. While the peak strength of TZN-3 is twice that of ZrC, it is not as strong as monolithic ZrC at T > 2000 °C.

**Figure 5 materials-16-06158-f005:**
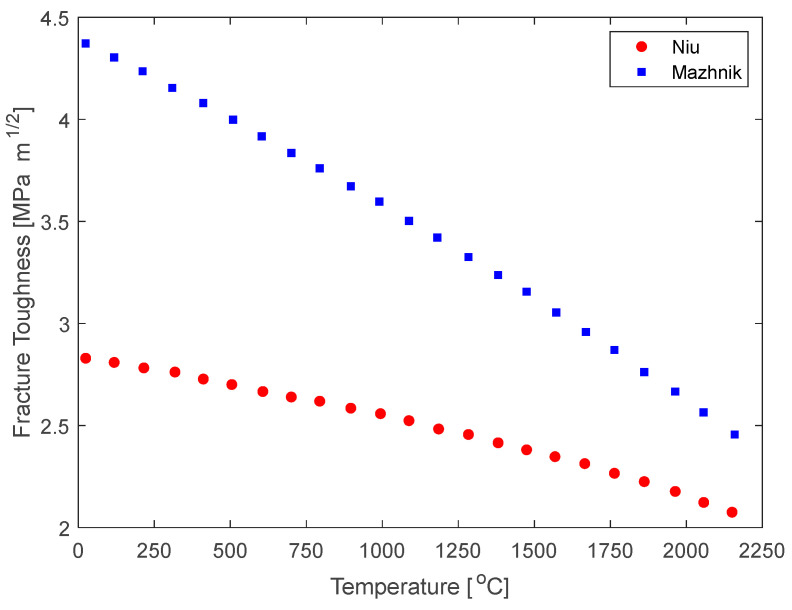
Fracture toughness vs. temperature calculated from First Principles Density Functional Theory. As ultra-high temperatures can present challenges, models aid in predicting performance and are useful to predict mechanical behaviors and properties in these extreme environments [38,39].

**Figure 6 materials-16-06158-f006:**
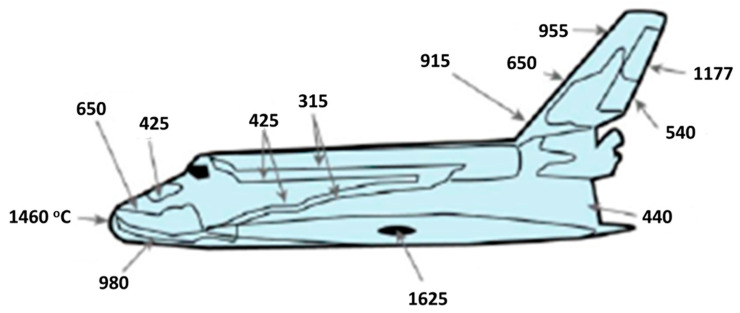
Nominal surface temperature profile of Space Shuttle Orbiter in degrees Celsius adopted from glass [14].

**Figure 7 materials-16-06158-f007:**
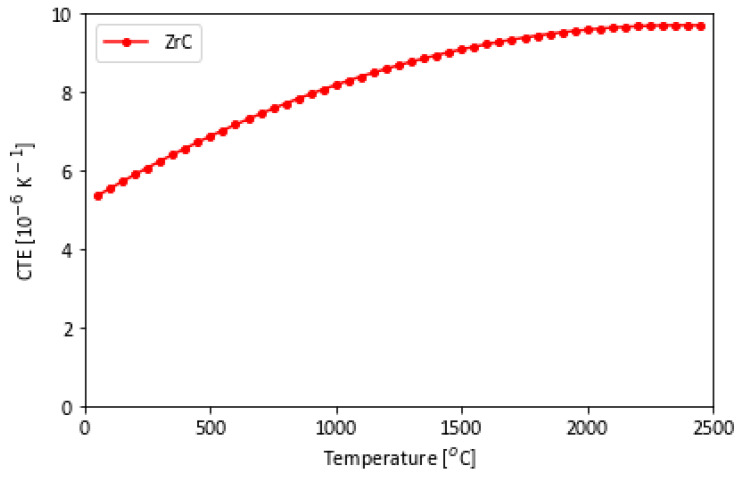
CTE vs. temperature for ZrC from measured values based on Touloukian’s compilation [15].

**Figure 8 materials-16-06158-f008:**
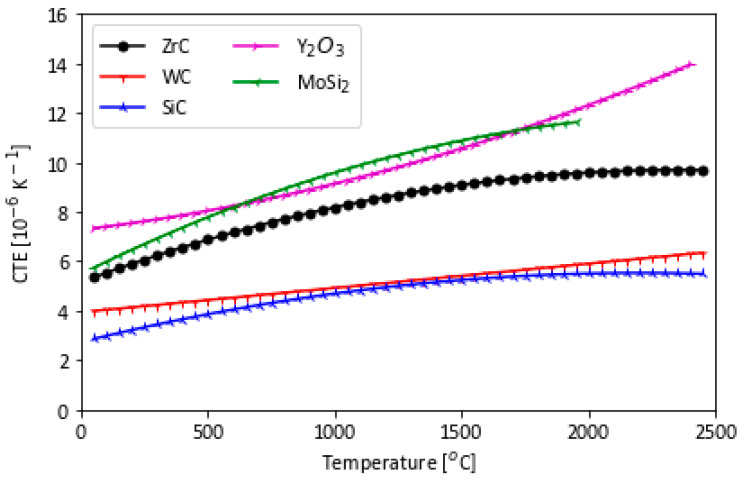
CTE vs. temperature for polycrystalline substrate candidates of WC, SiC, and ZrC with potential coating candidates of Y_2_O_3_ and MoSi_2_ are shown. ZrC offers a higher temperature range coupled with better CTE matching to potential coatings of MoSi_2_ and Y_2_O_3_ compared with other ceramics such as WC or SiC, which have a greater CTE mismatch with Y_2_O_3_ and MoSi_2_ [15].

**Figure 9 materials-16-06158-f009:**
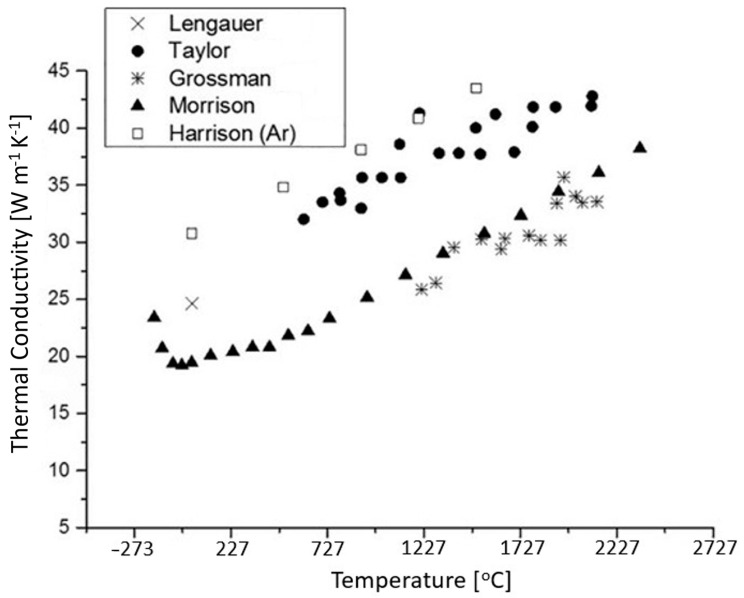
Thermal conductivity of ZrC vs. temperature. While there is considerable variance in measurements as noted by the authors, there is a common trend of increasing thermal conductivity that is atypical of ceramics and is attributed to the metal-like behavior of the Zr-Zr bonds at high temperatures [4].

**Figure 10 materials-16-06158-f010:**
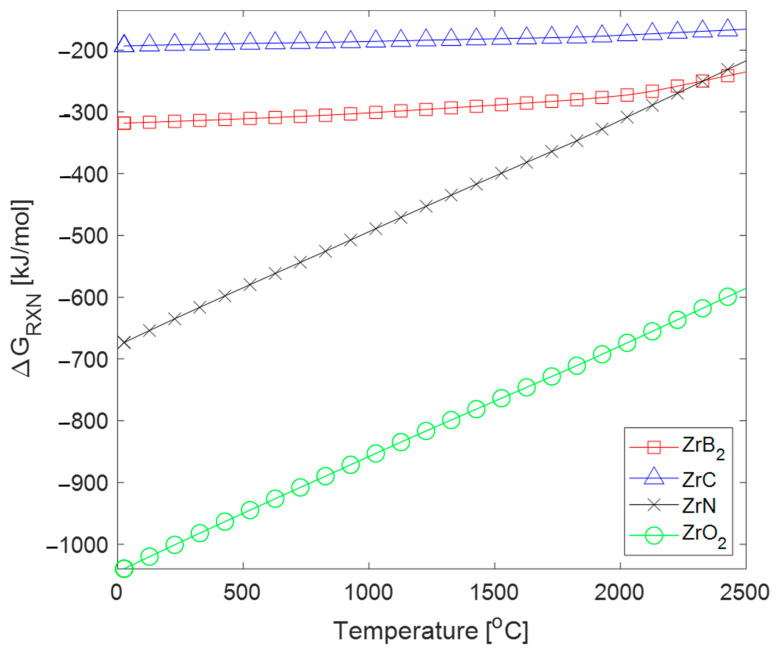
Gibbs energy of reaction vs. temperature of Zr and nonmetallic species (C, B, N_2_, and O_2_) for high-temperature Zr compounds [48].

**Figure 11 materials-16-06158-f011:**
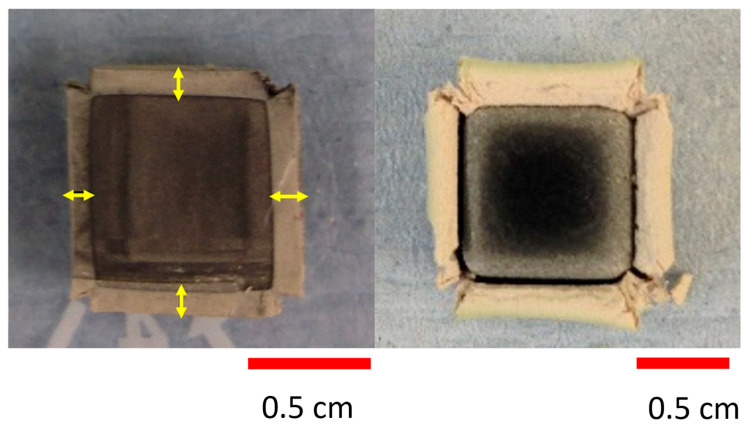
Oxide Layer Growth on ZrC (Left) at 800 °C for 4 h and (Right) at 1100 °C for 30 min [18].

**Figure 12 materials-16-06158-f012:**
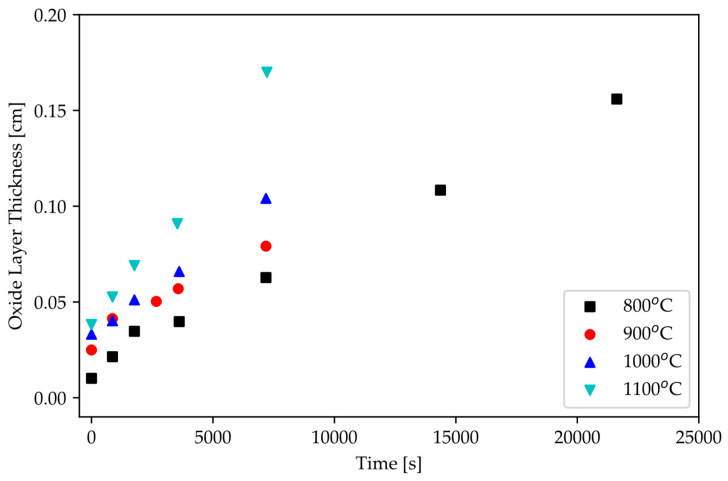
Oxide scale growth on ZrC from 800 °C to 1100 °C in air [17].

**Figure 13 materials-16-06158-f013:**
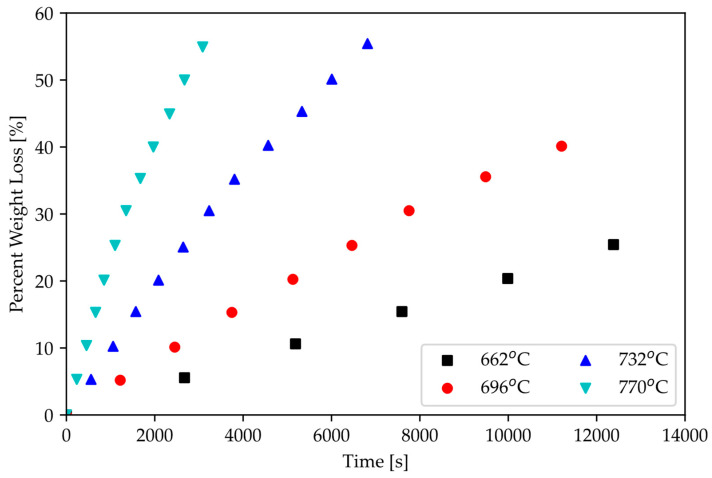
Mass loss due to oxidation vs. time and temperature of C/C composites [19].

**Figure 14 materials-16-06158-f014:**
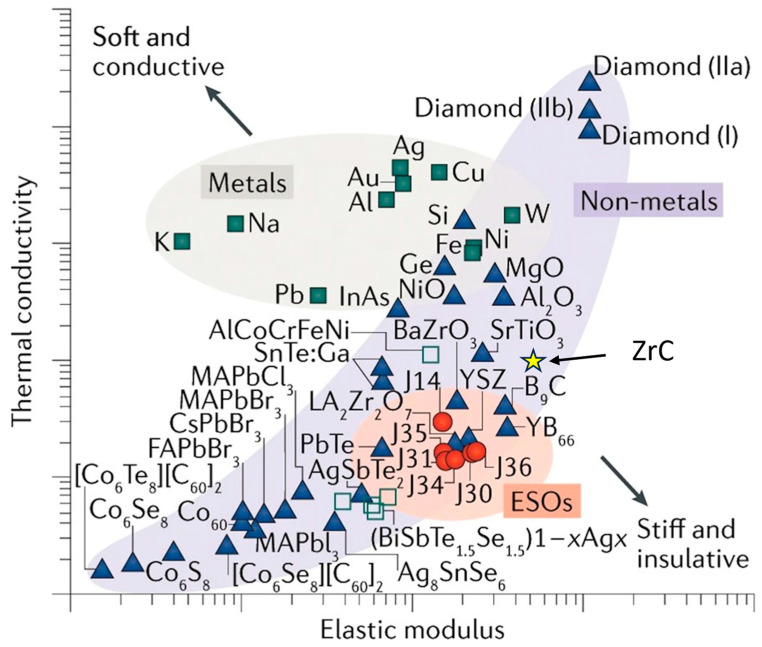
Ashby plot of thermal conductivity vs. elastic modulus in log–log scale. The corresponding values for ZrC are shown [51].

**Figure 15 materials-16-06158-f015:**
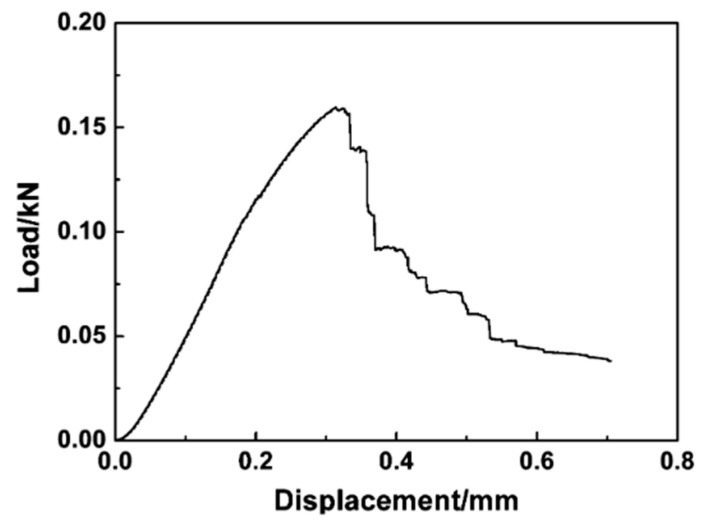
Load vs. displacement for Type-B, 3-pt flexure test of C/C–ZrC–ZrB_2_ showing the effects of forming a composite to prevent abrupt failure as indicative of the cascading failure beyond 0.3 mm [53].

**Figure 16 materials-16-06158-f016:**
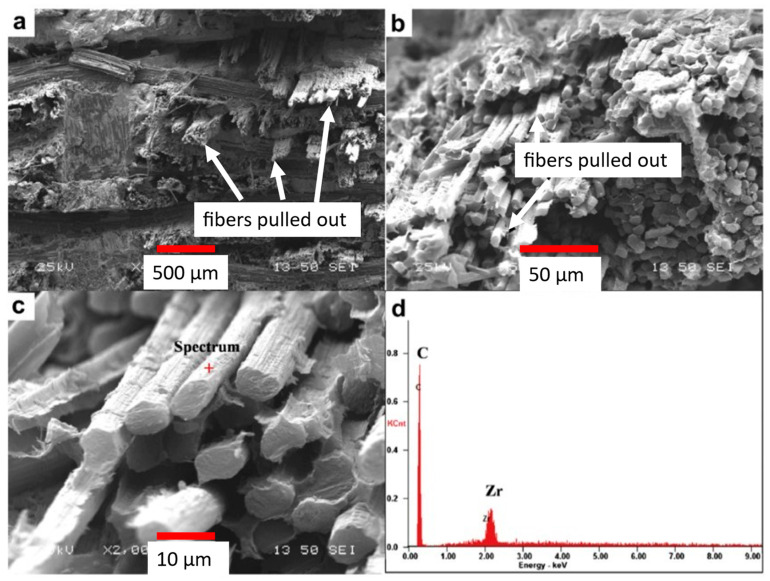
The fracture surface of ZrB2 reinforced with C/C Fibers (**a**). Evidence of C/C fiber pullout at the fracture surface (**b**). Fiber fracture (**c**), EDS results from fiber surface (**d**) [53].

**Figure 17 materials-16-06158-f017:**
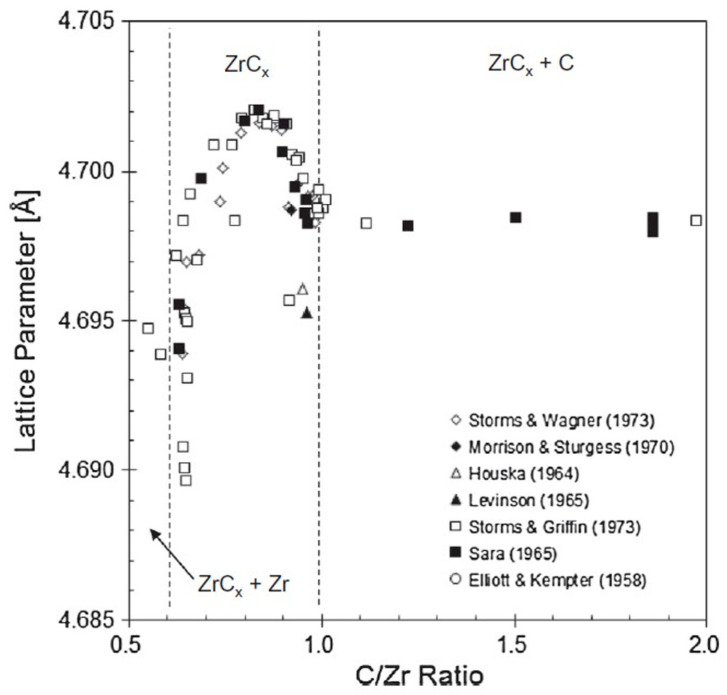
C/Zr ratio effect on ZrC lattice parameter [3].

**Figure 18 materials-16-06158-f018:**
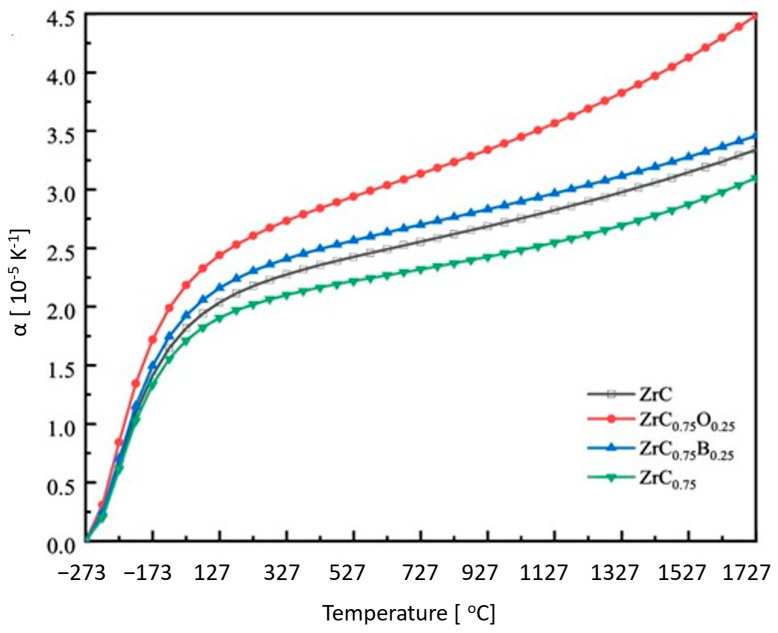
(**Top**) CTE and (**Bottom**) volumetric change vs. temperature of doped and undoped ZrC [54].

**Figure 19 materials-16-06158-f019:**
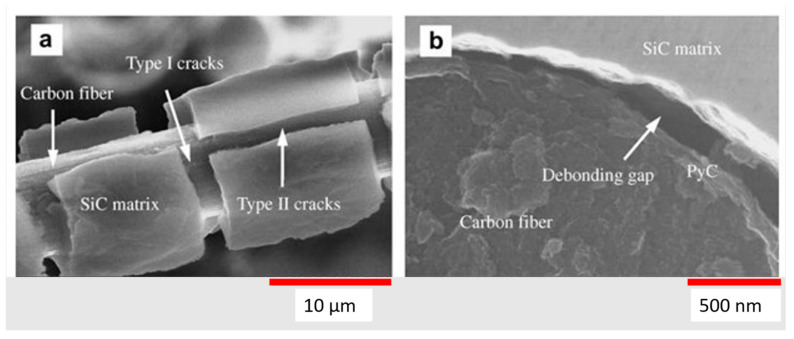
Example of cracks from thermal processing of C/SiC. Cracks form parallel (Type II) and perpendicular (Type I) to the carbon fiber (**a**). Debonding between carbon fiber SiC matrix (**b**) [45].

**Figure 20 materials-16-06158-f020:**
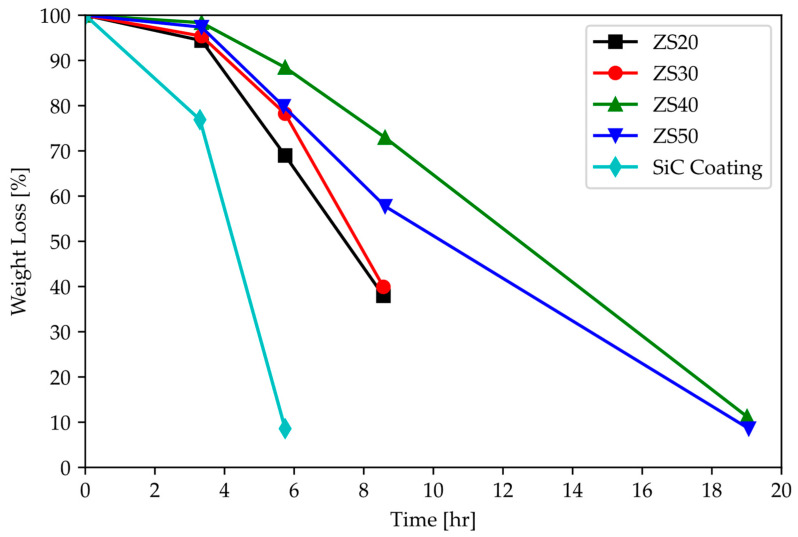
(**Top**) Percent weight loss vs. time at 1500 °C and (**Bottom**) percent weight loss during thermal cycling from room temperature to 1500 °C for ZrB_2_-SiC/SiC and SiC for graphite-coated specimens. From ZS20 to ZS50, the amount of ZrB_2_ is decreased, and SiC is increased from 43% to 71%. The coatings with the higher SiC phase fraction performed better. The ZS40 and ZS50 coatings performed similarly at early and late times, but ZS40 performs better at intermediate times. For WLE applications, the hold time is less important than cycling. Lastly, the SiC coating is the weakest performer in both cases [56].

**Figure 21 materials-16-06158-f021:**
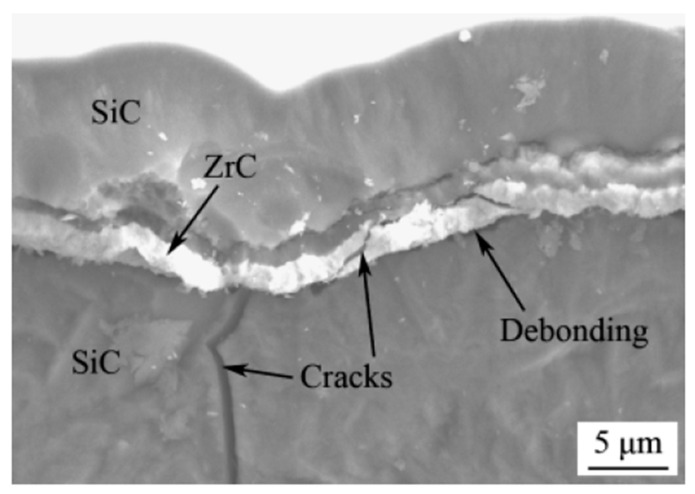
Layer formation during coating with SiC-ZrC of SiC [45].

**Figure 22 materials-16-06158-f022:**
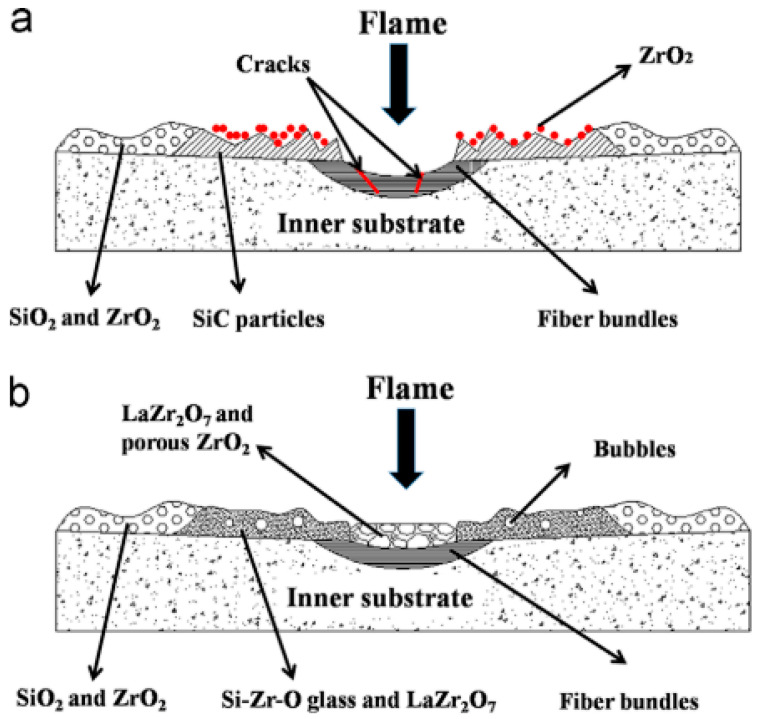
(**a**) C/SiC-ZrC-Y_2_O_3_ coating and (**b**) C/SiC-ZrC-La_2_O_3_ coating ablation mechanism [60].

**Table 1 materials-16-06158-t001:** Material properties of C/C and ZrC.

Property	C/C	ZrC
Melting Point (°C)	3552	3532
CTE (K^−1^)	−0.6	4
Density (g cm^−3^)	1.3 to 2.5	6.73
Thermal Conductivity (W m^−1^ K^−1^)	7.5–47.5	17.5–31.5
Flexure Strength (MPa)	140 ± 8	460 ± 24
Elastic Modulus (GPa)	43 to 240	435

**Table 2 materials-16-06158-t002:** ZrC mechanical, thermal, and chemical properties of interest for WLE applications.

Mechanical	Thermal	Chemical Interactions
**Strength (Elastic, Compressive, Failure, Tensile)**	CTE	Oxidation Resistance
**Fracture Toughness**	Thermal Shock	Coating/Alloying Considerations (Chemical Compatibility)
**Density**	Melting Temperature	
	Thermal Conductivity	

**Table 3 materials-16-06158-t003:** The room temperature moduli of ZrC [35,36].

	C_11_	C_12_	C_44_	Poisson Ratio	Source
**DFT**	504–514	90–112	157–173	0.18	[35]
**Experimental**	470	100	160	0.18	[36]

**Table 4 materials-16-06158-t004:** Comparison of mechanical properties for ZrC and TZN-3 [11,32,37].

Hardness (GPa)	Young’s Modulus (GPa)	Flexure Strength RT (MPa)	Flexure Strength 1600 °C (MPa)	Flexure Strength 1800 °C (MPa)	Fracture Toughness (MPa m^1/2^)
**TZN-3** **25–28**	420–470	460 ± 24	496 ± 44	366 ± 46	2.9
**ZrC**	435	192 ± 20	192 ± 20	310 ± 63 (2000 °C)	193 ± 110 (2500 °C)

**Table 5 materials-16-06158-t005:** Experimental values of ZrC with the addition of MoSi_2_ as a sintering aid [28].

Initial Composition (Vol %)	Sintering Parameters (°C/min/MPa)	Mean Grain Size (μm)	Density (g cm^3^)	Nanoindentation Hardness (GPa)	Elastic Modulus (GPa)	Vickers Hardness (HV1.0)	Fracture Toughness, K_ic_ (MPa m^1/2^)	3-pt Flexure Test (MPa)
ZrC	2100/3/65	13 ± 1	10.1	25.2 ± 1.4	464 ± 22	17.9 ± 0.6	-	407 ± 38
ZrC + 1MoSi_2_	1950/3/100	7.0 ± 0.7	12.3	24.9 ± 1.0	420 ± 10	18.8 ± 0.3	2.1 ± 0.2	495 ±61
ZrC + 3MoSi_2_	1900/3/100	5.8 ± 0.6	12.4	25.1 ± 2.3	444 ± 17	18.4 ± 0.7	3.2 ± 0.4	-
ZrC + 9MoSi_2_	1700/3/100	3.5 ± 0.4	11.9	26.8 ± 1.3	467 ± 22	20.0 ± 0.5	3.3 ± 0.4	591 ± 48

**Table 6 materials-16-06158-t006:** Material properties for ZrC vs. processing conditions at two temperatures [27].

Sintered Ceramics	Sintering Temperature (°C)	Real Density (g cm^−3^)	Theoretical Density (g cm^−3^)	Relative Density (%)	Mean ZrC Grain Size (micron)	Fracture Strength (MPa)	Vickers Hardness (GPa)
As-received ZrC	1900	5.46	6.7	81.5	-	-	-
MZ	1900	5.8	6.7	86.6	1.15	-	-
ZS	1900	5.45	6.0	90.8	1.56	-	-
ZC	1900	6.37	6.7	95	6.94	327.6 ± 9.1	9.4 ± 0.4
As-received ZrC	2100	6.33	6.7	94.4	18.59	337.6 ± 8.3	8.9 ± 0.4
MZ	2100	6.59	6.7	98.4	5.97	349.7 ± 15.0	10.8 ± 1.1
ZS	2100	5.8	6.0	96.7	3.08	473.5 ± 0.1	11.8 ± 0.8
ZC	2100	6.31	6.7	94.1	15.72	-	-

**Table 7 materials-16-06158-t007:** Novel enhancements to improve ZrC for WLE applications.

Enhancement	Advantages	Disadvantages
**Composites**	Increase ductility Increase fracture toughness	Increase CTE mismatch considerations Lowering the oxidation resistance
**Sintering aids**	Increase relative density Increase strength	Increase CTE mismatch considerations Increase defect density Lower the oxidation resistance
**Dopants Thermal Property Enhancement**	Increase CTE Increase relative density	Lower the oxidation resistance Increase defect density
**Dopants Chemical Property Enhancement**	Increase stability at high temperatures	Increase defect density Decrease Thermal conductivity
**Coatings**	Prevent oxidation	Limited to the strength, temperature, and permeability of coating

## Data Availability

Data sharing not applicable.

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
