# Peer review of "Zirconium Carbide for Hypersonic Applications, Opportunities and Challenges"

_materials, 2023, doi:10.3390/ma16186158_

Round 1
Reviewer 1 Report
materials-2573573
ZIRCONIUM CARBIDE FOR HYPERSONIC APPLICATIONS, OPPORTUNITIES AND CHALLENGES
Comments: After careful evaluation, I have concluded that the above-mentioned manuscript requires major revision.
- How does the unique bonding pattern between Zr-Zr and Zr-C atoms in the FCC, rock salt structure of ZrC affect its mechanical and thermal properties, particularly at high temperatures?
- Could variations in the Zr:C ratio impact the mechanical, thermal, and chemical properties of ZrC? How might this ratio be optimized for specific applications?
- What are the potential implications of lattice distortions caused by vacancies and defects on the mechanical properties of ZrC? How can these effects be mitigated or controlled?
- Can authors elaborate on the mechanisms that lead to the increase in thermal conductivity of ZrC at higher temperatures, despite the typical behavior of ceramics to exhibit decreasing thermal conductivity?
- Given the challenges associated with ZrC sintering, can authors provide further insights into the role of sintering aids, like MoSi2, in enhancing densification and improving mechanical properties?
- How does the introduction of carbon/carbon (C/C) composites into ZrC impact its overall material properties, and how can the trade-offs between properties be optimized for specific aerospace applications?
- Considering the importance of dimensional stability and CTE matching, could authors discuss potential strategies to prevent micro-cracking and delamination at the interface between ZrC and protective overcoats?
- What are the key mechanisms responsible for the decrease in ZrC flexure strength at temperatures above 2400°C? How might this impact the material's performance in hypersonic environments?
- Can authors provide insights into the fundamental reasons behind the decrease in flexure strength of ZrC at temperatures higher than 1600°C, as observed in the case of (Ta,Zr,Nb)C ceramic?
- How might the presence of carbon deficiency influence defect diffusion, mechanical behavior, and oxidation resistance of ZrC at elevated temperatures?
- In the context of hypersonic applications, what are the most critical challenges in terms of corrosion control for ZrC? How could doping, alloying, or coating approaches be employed to address these challenges?
- Considering the volumetric changes experienced by ZrC during intermediate temperatures of hypersonic flight, what strategies could be explored to minimize these changes and enhance the compatibility of ZrC with composites or coatings?
- Could authors discuss the potential role of ZrC in other components of hypersonic platforms beyond wing leading edges, based on its unique properties and stability?
- How might the mechanical behavior of C/C-ZrC composites differ from pure ZrC and C/C materials? Are there potential challenges or synergies arising from their combined use?
- Can authors provide examples of other high-temperature environments, beyond hypersonic applications, where ZrC's properties could be leveraged effectively?
Reviewer 2 Report
This review paper introduced Zirconium Carbide as an excellent material candidate for use in hypersonic areas. To be detailed, they chose C/C-ZrC composite as the solution for the WLE component and explained its advantages, such as the addition of C/C lowers the overall density compared to pure ZrC, and promotes nonbrittle fracture as well as increases the ductility of the materials.
For the Part I Introduction part, the authors explained ZrC’s physical properties, crystal structure, sintering parameters, and material properties for WLE including its mechanical properties. However, all the equations, L175-278 P6, should be numbered accordingly. It is suggested that the resolution of some figures (like Figs. 6, 7, 9, 11, 13, and 18) should be improved to meet the qualification requirement.
Also, the figure in part 2.2.3 (page 10) was mistakenly labeled as figure 1 (it should be figure 8).
In Table 7, the disadvantages should be in more detail, such as lowering the oxidation resistance instead of using “effect”.
The unit of temperature should be kept consistent. The authors used both C and K.
Consider rephrasing the sentence "Fracture toughness is a deficiency of ceramics, so the ability to limit brittle failure is very beneficial." for better clarity.
In the "Future Work" section, when discussing corrosion control and volumetric changes, it would be valuable to elaborate on how these challenges can impact the performance of C/C-ZrC composites in practical applications.
Some sentences could be restructured for better clarity and coherence. For example, in the sentence "Herein lies the next two major challenges: corrosion control and volumetric changes,"
There are some minor typographical errors and formatting issues that need to be corrected to enhance the professionalism of the manuscript.
Reviewer 3 Report
The paper “Zirconium Carbide for Hypersonic Applications, Opportunities and Challenges” is devoted to preparation and investigation of the mechanical, thermal, and chemical properties of the ZrC materials. This review covers various methods of sintering of the ZrC materials. The results of investigation of the thermal conductivity, coefficient of thermal expansion, dimensional stability at high temperatures, thermodynamic stability, and reaction kinetics are discussed in details. The work is of scientific interest to specialists in the field of obtaining new materials for aerospace applications. The data are reliable and do not cause much doubt. Nevertheless, there are several points before the paper can be published. I hope that authors after major revisions can improve the paper and can publish it in Materials.
- Introduction part should be improved by new relevant literature in the field of obtaining materials based on elements with a high melting point for aerospace application. I suggest to use the following references (see and discuss:
https://doi.org/10.3390/nano10020370;
https://doi.org/10.3390/nano12101642).
- The main idea of the presented review should be highlighted in the Introduction.
- Some figures have a very bad resolution (figs. 1, 7-9, 11-14, 17,18, 20 and 22). I suggest to rebuild the graphs and re-draw the images. Now they it is very difficult to evaluate them.
- Why did you not to discuss the crystal structure of the ZrC materials. I think that it is very important to investigate the correlation of the sintering parameters of materials and crystal structure.
- What practical recommendation can you suggest to the readers in term of these materials’ application?
- Conclusion part should be inserted to the manuscript.
Round 2
Reviewer 1 Report
the revised manuscript can be accepted.
Reviewer 3 Report
Now the paper can be accepted in its present form.